# ResBit: Residual Bit Vector for Categorical Values

## Abstract

The one-hot vector has long been widely used in machine learning as a simple and generic method for representing discrete data. However, this method increases the number of dimensions linearly with the categorical data to be represented, which is problematic from the viewpoint of spatial computational complexity in deep learning, which requires a large amount of data. Recently, Analog Bits, a method for representing discrete data as a sequence of bits, was proposed on the basis of the high expressiveness of diffusion models. However, since the number of category types to be represented in a generation task is not necessarily at a power of two, there is a discrepancy between the range that Analog Bits can represent and the range represented as category data. If such a value is generated, the problem is that the original category value cannot be restored. To address this issue, we propose **Res**idual **Bit** Vector (ResBit), which is a hierarchical bit representation. Although it is a general-purpose representation method, in this paper, we treat it as numerical data and show that it can be used as an extension of Analog Bits using Table Residual Bit Diffusion (TRBD), which is incorporated into TabDDPM, a tabular data generation method. We experimentally confirmed that TRBD can generate diverse and high-quality data from small-scale table data to table data containing diverse category values faster than TabDDPM. Furthermore, we show that ResBit can also serve as an alternative to the one-hot vector by utilizing ResBit for conditioning in GANs and as a label expression in image classification.

## 1 Introduction

Generative Adversarial Networks (GANs) (Goodfellow et al., 2014), one of the deep generative models, was proposed in the field of image generation, but it has been applied to other fields as well (Yu et al., 2017; Guo et al., 2018; Pei et al., 2021; Jeha et al., 2022; Donahue et al., 2018; Engel et al., 2019). Recently, however, diffusion models (Sohl-Dickstein et al., 2015; Ho et al., 2020; Song et al., 2021) outperform GANs in the field of image generation (Dhariwal & Nichol, 2021; Saharia et al., 2022; Ramesh et al., 2022; Balaji et al., 2023; Xue et al., 2023), but they have been successful for other modalities (Austin et al., 2021; Li et al., 2022a; Chen et al., 2021; Kong et al., 2021; Li et al., 2022b; Vignac et al., 2023; Shabani et al., 2023).

Tabular data generation is no exception, and models using GANs such as TGAN (Xu & Veeramachaneni, 2018) and CTGAN (Xu et al., 2019) have been proposed. Moreover, models using diffusion models have also been proposed (Kotelnikov et al., 2023; Lee et al., 2023; Kim et al., 2023). In these methods, models are trained by converting categorical data to a one-hot vector. The one-hot vector is very widely used because of its high versatility and simplicity. However, as the type of categorical values to be represented increases, the dimensions increase linearly, and the space complexity increases accordingly. In addition, in our experiments that motivated us to begin this study, increasing the dimensionality can cause model learning to fail. To address these issues, methods using label encoding or sparse matrix are used in data analysis, but there are few studies using such methods in generative modeling. On the other hand, Analog Bits (Chen et al., 2023) decreases the dimensions by representing discrete values as binary bits. Also, on the basis of the high expressiveness of diffusion models, by treating Analog Bits as numerical data, generative models can generate discrete data without the need to devise a model for discrete data generation. While this method is very strong, it places limits on the values that can be represented in simple binaries. If the range of values is predetermined, such as the pixel values of an image, there is no problem. However, the categorical

values of tabular data vary depending on the training data, and binary bits represent even extra parts. One possible solution is to limit the number of categorical values in advance to the number that can be represented in binary bits (FUCHI et al., 2023), but this approach sacrifices the original diversity of the training data and is not a fundamental solution.

In this paper, we propose **Res**idual **Bit** Vector (ResBit), which is fusion of the idea of Analog Bits and Residual Vector Quantization (Juang & Gray, 1982). This is a method of representing data in a hierarchical structure, which can deal with the problems of the linear increase of the dimensionality in the one-hot vector and the extra expressiveness in binary bits. ResBit can be used not only for deep learning but also machine learning in general, but in our experiments, we confirm its effectiveness using Table Residual Bit Diffusion (TRBD), which incorporates Residual Analog Bits that treat ResBit as numerical data into TabDDPM (Kotelnikov et al., 2023) as a motivating use case. We also show that ResBit can be used as a conditioning method by incorporating it into CGAN (Mirza & Osindero, 2014) to generate class-specified images. Finally, we show that ResBit can also be used in image classification tasks. In summary, our contributions are as follows:

- We propose ResBit, which is a novel method for representing discrete/categorical data.

- We propose TRBD, which is incorporated into TabDDPM by treating ResBit as numerical data. Experiments show that TRBD performs as well as TabDDPM and can be trained and generated at higher speed.

- We experimentally confirm that ResBit can be used for conditioning and labeling as an alternative to the one-hot vector.

## 2 RELATED WORKS AND PRELIMINARIES

### 2.1 DIFFUSION MODELS

Diffusion models (Sohl-Dickstein et al., 2015) are one of the deep generative models defined from forward and reverse Markov processes. In the forward process, real data $\boldsymbol{x}_0$ is added noise until it approximates pure noise $\boldsymbol{x}_T$. This process is represented as $q(\boldsymbol{x}_t|\boldsymbol{x}_{t-1}) = \mathcal{N}(\boldsymbol{x}_t; \sqrt{1-\beta_t}\boldsymbol{x}_{t-1}, \beta_t\boldsymbol{I})$, where $\mathcal{N}(\mu, \Sigma)$ denotes a Gaussian distribution with mean $\mu$ and variance $\Sigma$ and $\beta_t \in \mathbb{R}$ determines the strength of added noise and satisfies $0 < \beta_1 < \beta_2 < \ldots < \beta_T < 1$. The model trains the reverse process represented as $p_\theta(\boldsymbol{x}_{t-1}|\boldsymbol{x}_t) = \mathcal{N}(\boldsymbol{x}_{t-1}|\mu_\theta(\boldsymbol{x}_t, t), \Sigma_\theta(\boldsymbol{x}_t, t))$. In Denoising Diffusion Probabilistic Models (DDPM) (Ho et al., 2020), variance is fixed as $\sigma_t\boldsymbol{I}$ where $\alpha_t = \frac{1-\overline{\alpha}_{t-1}}{1-\overline{\alpha}_t}\beta_t$ with $\overline{\alpha}_t = \Pi_{i=1}^t \alpha_i$ and $\alpha_i = 1 - \beta_i$. The model is trained to maximize the variational lower-bound on $\log p_\theta$ and equation 1 is used as the simplified objective function.

$$L_t^{\text{simple}} = \mathbb{E}_{t, \boldsymbol{x}_0, \boldsymbol{\varepsilon}}[\|\boldsymbol{\varepsilon} - \boldsymbol{\varepsilon}_\theta(\boldsymbol{x}_t, t)\|^2] \tag{1}$$

Here, $\boldsymbol{\varepsilon}_\theta$ denotes the neural network and predicts added noise $\varepsilon \sim \mathcal{N}(0, \boldsymbol{I})$.

### 2.2 TABDDPM

To our knowledge, TabDDPM (Kotelnikov et al., 2023) is the first study in the world to introduce diffusion models to tabular data generation and outperforms existing deep generative models for tabular data (Xu et al., 2019; Xu & Veeramachaneni, 2018; Zhao et al., 2021) in terms of quality of sampling. TabDDPM consists of Gaussian diffusion models for numerical data and multinomial diffusion models (Hoogeboom et al., 2021) for categorical ones, and it uses equation 2 as the objective function for learning.

$$L_t^{\text{TabDDPM}} = L_t^{\text{simple}} + \frac{\sum_{i \leq C} L_t^i}{C} \tag{2}$$

Here, $C$ denotes the number of categorical features. The predefined distributions for categorical data are represented as $q(\boldsymbol{x}_t|\boldsymbol{x}_{t-1}) = \text{Cat}(\boldsymbol{x}_t; (1-\beta_t)\boldsymbol{x}_{t-1} + \beta_t/K)$ when the number of classes is $K$.

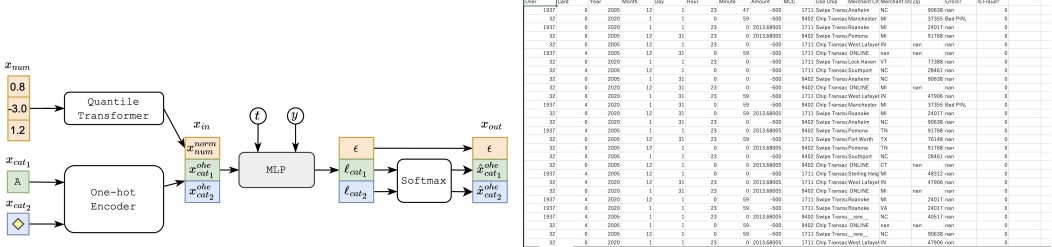

Figure 1: TabDDPM scheme. This is cited from Kotelnikov et al. (2023)

Figure 2: Example of failure case

Figure 1 shows the scheme of TabDDPM. The preprocessing of TabDDPM is divided into two parts: numerical and categorical data, which are preprocessed by Quantile Transformer and One-Hot Encoder, respectively. The preprocessed data is the input to the model. Table 1 shows whether a mode collapse-like phenomenon, as shown Figure 2, occurred when we randomly sampled 55,000 cases from IBM's synthetic credit card data[1] and changed the number of unique values for the data of each category. The percentage here is the percentage of the total training data, and data with frequencies lower than this percentage are masked[2]. Although the dataset used here is only a partial selection, it contains a very large number of categorical values. In this experiment, the hyperparameters except the value indicating the masking ratio are fixed, so changes in this ratio are linked to changes in the dimensionality of the one-hot vector given as input to the multinomial diffusion models. On the basis of this experiment, we hypothesize that the increase in the dimensionality of the one-hot vector affects the success or failure of training.

Table 1: Success or failure when rate of masking was changed in experiment

| Threshold (Rate) | Success or Failure |
|---|---|
| 0.00038 | Success |
| 0.00028 | Failure |
| 0.00019 | Failure |
| 0.00010 | Failure |

## 2.3 ANALOG BITS

Analog Bits is a preprocessing method proposed for handling discrete data in Bit Diffusion (Chen et al., 2023). Normally, discrete data is represented as a one-hot vector, but Analog Bits represents it as bit strings. Furthermore, by treating bit strings as numerical data, continuous state diffusion models can be applied to discrete data without devising a specific architecture for discrete data.

## 2.4 RESIDUAL VECTOR QUANTIZATION

Vector Quantization is the process of approximating a set of data represented by a vector into a finite number of representative patterns. Residual Vector Quantization (RVQ) (Juang & Gray, 1982) represents the original vector as a hierarchical structure. This enables high precision quantization with small errors from the original vector and reduce the size of the set of codes required for quantization.

## 2.5 CONDITIONAL GAN

Conditional GAN (CGAN) (Mirza & Osindero, 2014) is a method to control the generation in GANs by conditioning. The objective function is represented as equation 3, where $x$, $y$ and $z$ denote image, conditional information, and noise respectively. For example, in image generation, conditional

---

[1] https://ibm.ent.box.com/v/tabformer-data
[2] This is represented as cat_min_frequency in the official TabDDPM implementation.

information is connected to the input image or noise to be input to two models, a generator and discriminator.

$$\min_G \max_D V(D, G) = \mathbb{E}_{\boldsymbol{x} \sim p(\boldsymbol{x})}[\log D(\boldsymbol{x}|\boldsymbol{y})] + \mathbb{E}_{\boldsymbol{z} \sim p(\boldsymbol{z})}[\log(1 - D(G(\boldsymbol{z}|\boldsymbol{y})))] \tag{3}$$

## 3 PROPOSED METHOD AND APPLICATION EXAMPLE

In this section, we introduce Residual Bit Vector (ResBit) and an application example, Table Residual Bit Diffusion (TRBD), which incorporates ResBit into TabDDPM.

### 3.1 OUT OF INDEX

We consider representing categorical values as binary bits. Unlike the one-hot vector, binary bits can limit the increase in dimensionality to a logarithmic increase, but it may produce values that cannot be represented in the task of generating categorical values. As an example of this problem, we consider the representation of the states of the United States. There are 50 states in the U.S. If we use a one-hot vector, state $j$ is represented as a 50-dimensional vector $\boldsymbol{e}^{(j)}$. On the other hand, in the binary bits case, 0 to 49 are represented as binary numbers $000000_{(2)}$ to $110001_{(2)}$. The dimension of this is 6. The generative model generates 0/1 for each element. Therefore, it is possible to generate $110010_{(2)} = 50$ to $111111_{(2)} = 63$, which is greater than 49 in these 6 dimensions. The values in this range cannot be converted to categorical values. This is an important challenge for this generation task. We call this problem **out of index** for convenience. A simple solution is to limit the number of categories in the training data by $2^n - 1$ for each column and mask the rest with a special string (FUCHI et al., 2023). In the example above, 31 states and the remaining number of states would be represented as 32 category values. However, this approach limits the number of category values and the model's ability to generate these values. Other possible solutions are to clip values outside the range or to correct values within the appropriate range by normalizing the values. However, the former causes a change in the distribution after generation, and the latter causes correctly generated values to be incorrect.

### 3.2 RESIDUAL BIT VECTOR

To overcome the out of index problem mentioned above, we propose **ResBit**, which is inspired by RVQ. In ResBit, we consider obtaining layered binary bits like RVQ. We consider an integer $M$ represented by ResBit. $M$ has an integer $b_1$ greater than or equal to zero and satisfies inequality 4. If $2^{b_1} - 1$ is represented as binary bits, all elements are 1, and its length is $b_1$.

$$2^{b_1} - 1 \le M < 2^{b_1+1} - 1 \tag{4}$$

In this state, only values between 0 to $2^{b_1} - 1$ can be represented, but the values between $2^{b_1}$ to $M$ cannot be represented. Next, we consider obtaining the binary bits of the difference, $M - (2^{b_1} - 1)$. In this case, there exists an integer $b_2$ greater than or equal to 0 that satisfies inequality 5.

$$2^{b_2} - 1 \le M - (2^{b_1} - 1) < 2^{b_2+1} - 1 \tag{5}$$

By repeating this operation, a set of several binary bits is obtained. If all bits in the set were 1, this would represent $M$, and therefore no value greater than $M$ can be represented in ResBit. This makes it possible to the avoid out of index problem. The overall algorithm consists of two parts: one part to find the length of each element of binary bits, i.e. $b_1, b_2, \cdots$, and the other part to find the ResBit of an integer $M$. The pseudo code for the former part is shown in Algorithm 1, and the latter part is shown in Algorithm 2.

**Algorithm 1** Get $b_1, b_2 \ldots$ Algorithm.

```
def get_length_ResBit(kind_of_cat: int):
    # get length of ResBit for unique category
        size kind_of_cat
    res = []

    # 0-index
    max_num = kind_of_cat - 1

    while True:
      bits = bin(max_num)[2:]

      # in case of M == 2^m
      if "0" not in bits:
        res.append(len(bits))
        break

      else:
        s = "1" * (len(bits) - 1)
        max_num -= int(s, base=2)
        res.append(len(s))

    return res
```

**Algorithm 2** Get ResBit Algorithm for $M$.

```
def int_to_ResBit(M, l):
    # l is returns of algorithm 1
    res = []
    for i in range(len(l)):
        if M == 0:
            res += [0 for _ in range(l[i])]
            continue

        bits = "1" * l[i]
        X = int(bits, base=2)
        if X <= M:
            res += [int(z) for z in bits]
            M -= X
        else:
            bits_M = bin(N)[2:]
            bits_M = "0" * (l[i] - len(bits_M)
                ) + bits_M
            res += [int(z) for z in bits_M]
            M = 0

    return res
```

At the end of this section, we reconsider an example of for representation of the states of the U.S. illustrated in Section 3.1. To represent the 50 states, Algorithm 1 gives $b_1 = 5, b_2 = 4, b_3 = 2$. Here, ResBit is given by:

$$\boldsymbol{x}_j = (\boldsymbol{y}_{[0,2^5)}, \boldsymbol{y}_{[0,2^4)}, \boldsymbol{y}_{[0,2^2)}). \tag{6}$$

When $j = 38 = 31 + 7$ for example, it is represented by:

$$\boldsymbol{x}_{38} = ((1,1,1,1,1),(0,1,1,1),(0,0)) = (1,1,1,1,1,0,1,1,1,0,0). \tag{7}$$

### 3.3 TRBD

In this section, we introduce Table Residual Bit Diffusion (TRBD) as an application case of ResBit. TRBD is based on the architecture of TabDDPM, but the preprocessing part for categorical data of TabDDPM was changed. Specifically, One-Hot Encoder is replaced by Residual Bit Encoder, and its outputs are treated as numerical data following the idea of Analog Bits. After converting each categorical value to its ResBit, the process described in equation 8 is performed. This is the same process as that of TabDDPM. Here, `ResBit_encoder` converts the category values to ResBit, and $x_{cat_i}$ is the $i$-th category column.

$$x_{cat_i}^{resbit} = \log(\max(\texttt{ResBit\_encoder}(x_{cat_i}), 10^{-30})) \tag{8}$$

The numerical data are combined with those obtained by equation 8, transformed by Quantile Transformer, and then input into the model.

$$x_{in} = \text{QuantileTransformer}(\text{concat}(x_{num}, x_{cat_1}^{resbit}, x_{cat_2}^{resbit}, \cdots)) \tag{9}$$

By performing this process, all data is treated as numerical data. At the time of generation, label representations are obtained by equation 10, and each one is converted to a category value as generated data. Here, `ResBit_decoder` converts ResBit to a categorical value.

$$x_{out}^{syn\_cat} = \texttt{ResBit\_decoder}(\text{round}(\exp(x_{out}^{resbit}))) \tag{10}$$

## 4 EXPERIMENTS AND EVALUATIONS

In this section, as an application example of ResBit, we conducted three types of experiments: tabular data generation, image generation, and image classification. All experiments were conducted

Table 2: Name of category column and dataset size for each dataset

| Dataset | Abbreviation | Categorical Column Name | Data Size |
|---|---|---|---|
| Credit Card | CC | Use Chip, Merchant City, Merchant State | 2.4M |
| Airlines | AR | Airline, Flight, AirportFrom, AirportTo | 600k |
| Insurance | IS | sex, smoker, region | 1338 |
| Buddy | BD | condition, color_type, X1, X2, pet_category | 18834 |
| Adult | AD | Workclass, Education, Marital_Status, Occupation, Relationship, Race, Sex, Native-Country | 48842 |

using a single Quadro RTX6000 GPU. The environments of the experiments used Ubuntu18.04, the CPU was an AMD EPYC 7502P 32-Core CPU at 2.5GHz. We used Python3.10.8, PyTorch1.13.1, CUDA10.1, and scikit-learn 1.1.2, but the versions sometimes differed for some experiments. In such cases, we report the version name.

## 4.1 TABULAR DATA GENERATION USING TRBD

### 4.1.1 DATASET

In this experiment, we selected 5 datasets, and they are listed in Table 2. For training TRBD, we used 600k pieces of training data for CC, 350k for AR, 856 for IS, 12,053 for BD, and 16,261 for AD. The sources of the data used in this experiment are given in Appendix D.

### 4.1.2 TRAINING PROCESS

Following Kotelnikov et al. (2023), we tuned hyperparameters by using Optuna (Akiba et al., 2019). Values evaluated by CatBoost (Prokhorenkova et al., 2018) were used for tuning. The tuning settings also conformed to those of Kotelnikov et al. (2023), but only the number of trials was changed to 30. The hyperparameter search spaces used throughout this experiment are shown in Appendix A.

### 4.1.3 EVALUATION METHODS

We used two metrics for evaluating TRBD.

**Number of Types of Categorical Values Generated** We compared the number of category values contained in the generated data for each category column. We confirmed by using this metric that the model can generate a wide variety of categorical data. The quality of the generated data was evaluated by the next metric.

**TSTR** Train on Synthesis, Test on Real (TSTR) (Esteban et al., 2017; Yoon et al., 2019) is one of the evaluation methods in tabular data generation that trains machine learning models with generated data and tests them with real data. Existing studies (Lee et al., 2023; Kim et al., 2023) use this method to measure the quality of sampling. We trained classifiers such as CatBoost and evaluated them using F1 and AUROC for classification tasks and $R^2$ and RMSE for regression tasks. We used 5 generated samples, evaluated each one in the TSTR framework, and report the means and standard deviations. We used scikit-learn 1.0.2 only for the training and generation phases of TRBD.

### 4.1.4 RESULTS AND DISCUSSION

Table 3 shows the results of TRBD evaluations using CatBoost. The number of generated samples was 300k for CC and AR and 10k for the others. The order of the number of categories generated in Table 3 corresponds to the order of the category column name in Table 2 from the left. The results for the TSTR framework are rounded to the 4th decimal place. The results using the other classifiers are shown in Appendix B. Regarding the TSTR framework, where we trained on synthetic data, we report the results of training on real data, which is referred to as Identity. We note here that TabDDPM tuning failed for CC and AR, so several cases were tested manually to show the best results. In the cases where tuning failed, the loss exploded or disappeared during the training phase, confirming that it was not backpropagated correctly. This is probably due to the very large number

Table 3: Result of experiments. For all metrics without RMSE, higher scores indicate better performance.

| Dataset | Methods | Number of category types | F1($R^2$) | AUROC(RMSE) |
|---------|---------|--------------------------|-----------|-------------|
| CC | Identity | (3, 7515, 153) | 0.684±0.067 | 0.971±0.009 |
|  | TabDDPM | (3, 3579, 153) | 0.000±0.000 | 0.376±0.075 |
|  | TRBD (ours) | **(3, 7513, 153)** | 0.000±0.000 | **0.746±0.051** |
| AR | Identity | (18, 6,571, 292, 293) | 0.587±0.004 | 0.727±0.004 |
|  | TabDDPM | (18, 5,369, 291, 291) | 0.279 ± 0.134 | 0.528 ± 0.013 |
|  | TRBD (ours) | **(18, 6,571, 292, 293)** | **0.482±0.010** | **0.638±0.003** |
| IS | Identity | (2, 2, 4) | 0.861±0.034 | 4,475.287±390.028 |
|  | TabDDPM | **(2, 2, 4)** | 0.894±0.003 | 4,050.754±47.229 |
|  | TRBD (ours) | **(2, 2, 4)** | **0.908±0.003** | **3,785.253±69.238** |
| BD | Identity | (3, 56, 19, 10, 4) | 0.987±0.000 | 0.931±0.001 |
|  | TabDDPM | (3, 49, 15, 9, 4) | **0.907±0.003** | **0.984±0.001** |
|  | TRBD (ours) | **(3, 55, 17, 9, 4)** | 0.901±0.007 | **0.983±0.001** |
| AD | Identity | (9, 16, 7, 15, 6, 5, 2, 42) | 0.716±0.003 | 0.927±0.002 |
|  | TabDDPM | **(9, 16, 7, 15, 6, 5, 2, 42)** | **0.673±0.002** | **0.909±0.003** |
|  | TRBD (ours) | **(9, 16, 7, 15, 6, 5, 2, 42)** | **0.676±0.004** | **0.906±0.000** |

Table 4: Comparison of training and sampling time

| Dataset | Method | input_dim | model_layer | train | sample | #params |
|---------|--------|-----------|-------------|-------|--------|---------|
| CH | TabDDPM | 16 | [256, 1024, 1024, 1024, 512] | 567s | 42s | 3M |
|  | TRBD (ours) | **12** | [512, 1024, 1024, 1024] | **264s** | **21s** | **2.7M** |
| IS | TabDDPM | 12 | [256, 512, 512, 512, 512, 256] | 407s | 30s | **1.1M** |
|  | TRBD (ours) | **8** | [1024, 512, 512, 1024] | **173s** | **17s** | 1.5M |

of dimensions. The input dimensions of CC were 7,676 in TabDDPM. On the other hand, TRBD succeeded in reducing the input dimensions to 75, which can be considered a successful learning process. Table 3 shows that AR, which had many category values for training data, could generate many types of category values compared with TabDDPM, and it outperformed in the quantitative evaluation. The results also show that TRBD is competitive with or outperforms TabDDPM not only in the case of large data but also small data. On the other hand, the F1 of CC does not show any performance at all. This may be related to the fact that CC is unbalanced data. The difference between CC and AR is whether it is balanced or unbalanced data, as the number of categorical values of CC is comparable to that of AR. The ratio of positive data to negative data, is almost 1:1 in AR and 0.001:0.999 in CC, which means that anomaly data with a high F1 score was not generated.

### 4.1.5 RUNTIME

TRBD is based on TabDDPM without the preprocessing part for category data. We confirmed how much it contributes to runtime. Runtime depends on the dataset and hyperparameters as well as TabDDPM. In this experiment, we used IS and the Churn Modeling (CH) dataset. We used $T = 1000$ and a batch size of 4096. The number of generated samples was 30k for IS and 26k for CH. Table 4 shows the results. By treating all data as numerical data, the input dimension was reduced, and the training and sampling time shortened simultaneously. The IS results also show that the model was faster even when the number of parameters increased, indicating that the multinomial diffusion part, which deals with categorical data, is a bottleneck in TabDDPM.

### 4.2 USE FOR CONDITIONING

ResBit also can be used for conditioning. In this section, we confirm this by using CGAN. We changed the conditioning part of CGAN from the one-hot vector to ResBit. Following the original

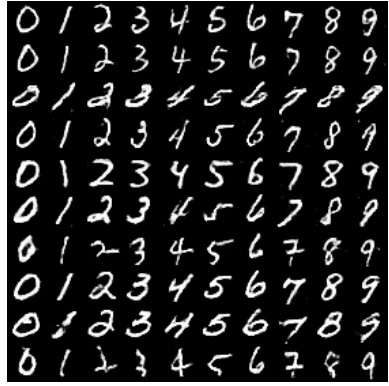

(a) One-hot conditioning

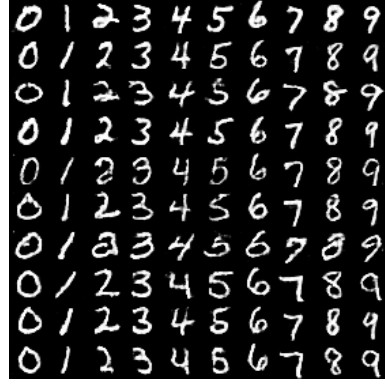

(b) ResBit conditioning (ours)

Figure 3: Class conditioned MNIST sample

Table 5: Results of two labeling methods on CIFAR-10

| Model | Labeling method | acc (%) |
|---|---|---|
| MobileNetV3-small | One-hot | 65.52 |
| | ResBit (ours) | 66.70 |
| MobileNetV3-large | One-hot | 74.63 |
| | ResBit (ours) | 72.55 |

CGAN work, a condition label is connected to a noise vector or image before each model input. Note that we do not use ResBit to represent pixels in an image because this has already experimented on and proven effective by Chen et al. (2023). We used the architecture of InfoGAN (Chen et al., 2016), whose noise dimension is 100. We experimented on the MNIST (LeCun et al., 2010), CIFAR-10 (Krizhevsky, 2009), and Food101 (Bossard et al., 2014) datasets and used the Adam optimizer (Kingma & Ba, 2017), whose hyperparameters were $\alpha = 0.0002, \beta_1 = 0, \beta_2 = 0.999$. For stable training, we used SNGAN (Miyato et al., 2018) on only the Food101 dataset. Since achieving quality with GANs is not the purpose of our paper, various methods of improving quality such as Heusel et al. (2017); Salimans et al. (2016); Gulrajani et al. (2017) are not incorporated. Figure 3 shows the samples generated using MNIST. One-hot and ResBit conditioning show that conditioning was successful. We show evaluations on CIFAR-10 and Food101 in Appendix C.

### 4.3 Use for Classification Labels

We showed that ResBit can be used for conditioning in Section 4.2. In addition, we also show that it can also be used as a label for classification tasks in this section. We used MobileNetV3 (Howard et al., 2017), a follow-up study to MobileNets (Howard et al., 2019), a lightweight, high-performance model intended for use in smartphones and other devices. The output of the model was treated similarly to multi-label classification. We used the CIFAR-10 dataset with a batch size of 64, Adam optimizer, and accuracy for the evaluation metric. We note that this experiment, like the GAN experiment, was also not designed to improve accuracy. Table 5 shows the results. Since the accuracy of ResBit corresponds to the total accuracy of multi-label classification, we consider the accuracy of ResBit to sometimes be lower than that of the one-hot method due to the increased difficulty of the task.

### 4.4 Reduction in Number of Dimensions

In Sections 4.2 and 4.3, we experimentally showed that ResBit can be used for conditioning and label representation with a limited number of classes. In this section, we compare the dimensionality of the one-hot vector and ResBit with increasing dimensionality. Figure 4 shows a comparison of the dimensionality of the one-hot vector and ResBit when the number of classes is increased. The

increase the dimensionality was linear for the one-hot vector, but the increase was very suppressed for ResBit. This is similar to what we confirmed with CC in Section 4.1.4. Combining this reduction with the uses of ResBit shown in Sections 4.2 and 4.3, ResBit can be used with a very small number of dimensions even if the number of classes increases. The demonstration of this is a subject for future work.

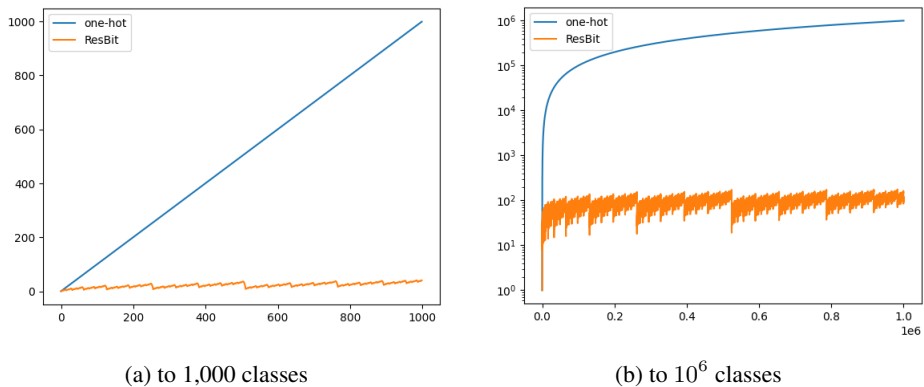

(a) to 1,000 classes $\qquad$ (b) to $10^6$ classes

Figure 4: Comparison of number of dimensions

## 5 CONCLUSION

In this paper, we proposed ResBit, which is inspired by Analog Bits and Residual Vector Quantization, as a method for representing discrete data. TRBD, which incorporates ResBit into TabDDPM and was our motivating use case was competitive or even superior in performance to TabDDPM, and it generated various categorical values and accelerated the speed of the process. Furthermore, we showed that ResBit can be used for conditioning and label representation by incorporating it into CGAN and MobileNetV3, demonstrating the high versatility of the proposed method. In addition, when the dimensionality of a one-hot vector is extremely large, ResBit can be used to dramatically reduce the dimensionality, which can be expected to shorten the training time in such a case. Future verification using other methods that use the one-hot vector is desirable.

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

## A  HYPERPARAMETERS SEARCH SPACE

We show the hyperparameter search spaces in Tables 6 and 7

Table 6: TRBD hyperparameter spaces from Kotelnikov et al. (2023). Only number of tuning trials is changed to 30.

| Parameters | Distribution |
|---|---|
| Learning rate | LogUniform[0.00001, 0.003] |
| Batch size | Cat{256, 4096} |
| Diffusion timesteps | Cat{100, 1000} |
| Training iterations | Cat{5000, 10000, 20000} |
| # MLP layers | Int{2, 4, 6, 8} |
| Width of MLP layers | Int{128, 256, 512, 1024} |
| Proportion of samples | Float{0.25, 0.5, 1, 2, 4, 8} |
| Number of tuning trials | 30 |

## B  ADDITIONAL RESULTS ON TSTR FRAMEWORK

We show the results using XGBoost (Chen & Guestrin, 2016) and RandomForest (Breiman, 2001) for the TSTR Framework. We used XGBoost library version 1.7.6 and scikit-learn 1.2.2. We note that this version of scikit-learn was used only in the TSTR evaluation stage, while 1.0.2 is used in the TRBD experiments. Tables 8 and 9 show the results using RandomForest and XGBoost, respectively.

Table 7: Hyperparameter space of classifier models from Kim et al. (2023); Lee et al. (2023) and Kotelnikov et al. (2023)

| Models | Parameters | Distribution |
|---|---|---|
| CatBoost | Learning rate | LogUniform[0.001, 1] |
| | Depth | UniformInt[3, 10] |
| | L2 leaf reg | Uniform[0.1, 10.0] |
| | Bagging temperature | Uniform[0, 1] |
| | Leaf estimation iterations | UniformInt[1, 10] |
| XGBoost | n_estimators | Int{10, 50, 100} |
| | min_child_weight | Int{1, 10} |
| | max_depth | Int{5, 10} |
| | gamma | Float{0.0, 1.0} |
| | nthreads | -1 |
| RandomForest | max_depth | Int{8, 16, Inf} |
| | min_samples_split | Int{2, 4} |
| | min_samples_leaf | Int{1, 3} |
| | n_jobs | -1 |
| Common settings | Iterations | 350 |
| | Early stopping rounds | 20 |
| | Number of tuning trials | 20 |

Table 8: Result of TSTR framework using RandomForest

| Dataset | Methods | F1($R^2$) | AUROC(RMSE) |
|---|---|---|---|
| CC | Identity | $0.597 \pm 0.040$ | $0.887 \pm 0.185$ |
| | TabDDPM | $0.000 \pm 0.000$ | $0.511 \pm 0.054$ |
| | TRBD (ours) | $0.000 \pm 0.000$ | $\mathbf{0.615 \pm 0.054}$ |
| AR | Identity | $0.572 \pm 0.003$ | $0.721 \pm 0.001$ |
| | TabDDPM | $0.136 \pm 0.087$ | $0.499 \pm 0.018$ |
| | TRBD (ours) | $\mathbf{0.479 \pm 0.002}$ | $\mathbf{0.637 \pm 0.002}$ |
| IS | Identity | $0.858 \pm 0.021$ | $4{,}737.314 \pm 330.580$ |
| | TabDDPM | $0.903 \pm 0.006$ | $3{,}880.795 \pm 115.701$ |
| | TRBD (ours) | $\mathbf{0.913 \pm 0.003}$ | $\mathbf{3{,}676.466 \pm 69.751}$ |
| BD | Identity | $0.930 \pm 0.004$ | $0.987 \pm 0.001$ |
| | TabDDPM | $\mathbf{0.790 \pm 0.013}$ | $\mathbf{0.978 \pm 0.002}$ |
| | TRBD (ours) | $0.751 \pm 0.050$ | $0.975 \pm 0.002$ |
| AD | Identity | $0.689 \pm 0.004$ | $0.916 \pm 0.002$ |
| | TabDDPM | $\mathbf{0.661 \pm 0.007}$ | $\mathbf{0.906 \pm 0.001}$ |
| | TRBD (ours) | $\mathbf{0.661 \pm 0.003}$ | $0.903 \pm 0.001$ |

## C  MORE RESULTS ON CGAN

We show the qualitative evaluation for CIFAR-10. We used Fréchet Inception Distance (FID) (Heusel et al., 2017) using the pytorch-fid (Seitzer, 2020) library. The FID results are shown in Table 10, and the generated samples are shown in Figure 5. There was no significant quality loss due to the change in the conditioning method. Next, we show the results for the Food101 dataset using SNGAN for stable training. Please see the paper in Miyato et al. (2018) for architecture details. For simplicity, we selected 40 classes out of 101 and created a dataset. Following the SNGAN experimental setup, we updated Discriminator 5 times for each Generator update. The number of training iterations was 15,000, and the batch size was 64. Generated examples are shown in Figure 6. The fact is that ResBit can be considered architecture-independent since it is able to generate the same results as before for both conditioning methods.

Table 9: Result of TSTR framework using XGBoost

| Dataset | Methods | F1($R^2$) | AUROC(RMSE) |
|---|---|---|---|
| CC | Identity | $0.620 \pm 0.066$ | $0.948 \pm 0.011$ |
| | TabDDPM | $0.000 \pm 0.000$ | $0.429 \pm 0.138$ |
| | TRBD (ours) | $0.000 \pm 0.000$ | $\mathbf{0.643 \pm 0.014}$ |
| AR | Identity | $0.595 \pm 0.002$ | $0.725 \pm 0.001$ |
| | TabDDPM | $0.353 \pm 0.095$ | $0.514 \pm 0.007$ |
| | TRBD (ours) | $\mathbf{0.485 \pm 0.004}$ | $\mathbf{0.619 \pm 0.002}$ |
| IS | Identity | $0.831 \pm 0.052$ | $4,777.171 \pm 690.880$ |
| | TabDDPM | $0.903 \pm 0.008$ | $3,872.860 \pm 156.814$ |
| | TRBD (ours) | $\mathbf{0.915 \pm 0.003}$ | $\mathbf{3,630.228 \pm 69.255}$ |
| BD | Identity | $0.927 \pm 0.005$ | $0.986 \pm 0.002$ |
| | TabDDPM | $\mathbf{0.573 \pm 0.002}$ | $\mathbf{0.935 \pm 0.007}$ |
| | TRBD (ours) | $0.568 \pm 0.002$ | $0.922 \pm 0.002$ |
| AD | Identity | $0.689 \pm 0.004$ | $0.926 \pm 0.002$ |
| | TabDDPM | $\mathbf{0.661 \pm 0.003}$ | $\mathbf{0.902 \pm 0.001}$ |
| | TRBD (ours) | $\mathbf{0.663 \pm 0.002}$ | $\mathbf{0.901 \pm 0.001}$ |

Table 10: Comparison on CIFAR-10 dataset

| conditioning method | one-hot | ResBit (ours) |
|---|---|---|
| FID | 95.52 | 82.21 |

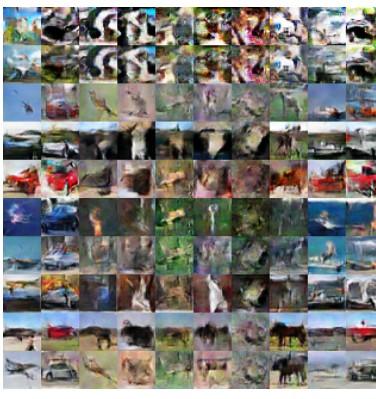
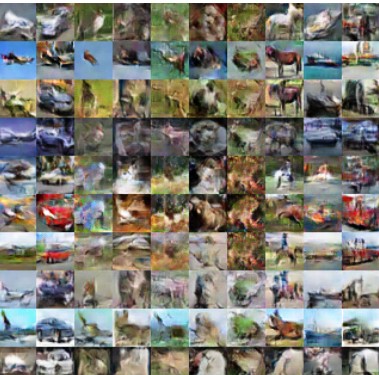

(a) One-hot conditioning        (b) ResBit conditioning (ours)

Figure 5: Class conditioned samples from InfoGAN trained on CIFAR-10

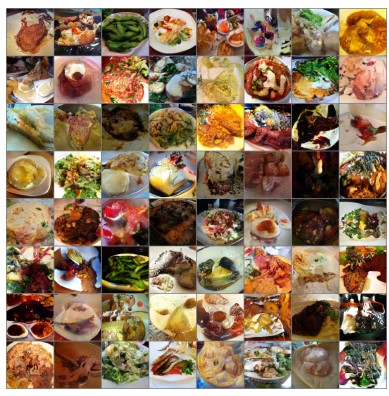 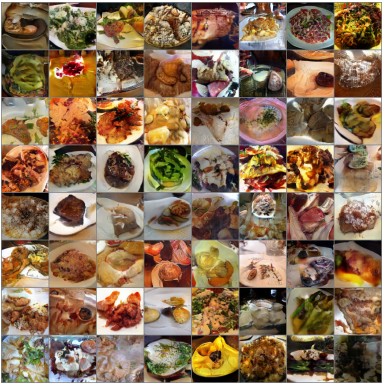

(a) One-hot conditioning         (b) ResBit conditioning (ours)

Figure 6: Generated samples from SNGAN trained on Food101

## D  TABULAR DATA SOURCES

- CC: `https://ibm.ent.box.com/v/tabformer-data`
- AR: `https://www.openml.org/search?type=data&sort=runs&id=1169&status=active`
- IS: `https://www.kaggle.com/datasets/mirichoi0218/insurance`
- BD: `https://www.kaggle.com/datasets/akash14/adopt-a-buddy`
- AD: Kohavi (1996)
- CH: `https://www.kaggle.com/datasets/shrutimechlearn/churn-modelling`

