# OpenReview forum: "ResBit: Residual Bit Vector for Categorical Values"
_ICLR.cc/2024/Conference — Submitted to ICLR 2024_

### Official Review · Reviewer_Zunf · 2023-11-01

**Soundness:** 3 good
**Presentation:** 2 fair
**Contribution:** 1 poor
**Rating:** 3
**Confidence:** 3

**Summary:**

The paper presents a new encoding technique for categorical values: residual bit vectors which are computed iteratively as bit-representations of category number (where category is treated as an actual number; see section 3.2. for a detailed explanation). The motivation for this work is in the application of one-hot encoded vectors: the authors are motivated to train table diffuison models where input/outputs are tabular data that can potentially have millions of categories. Training the diffusion model with million dimensional input outputs is a challenge, thus some lossless dimensionality reduction is needed. Why not compute bit representation once? Paper defines the main issue with simple bitwidth as an "out of index" problem, meaning that if total number of categories are not exact power of 2, say 9, then the bit representation of such a categories introduces extra "sampling" dimensions that might be an issue during diffusion training/sampling. In case of 9, its representation would require 4 bits, i.e. 9=1001; during diffusion sampling a number 1011 can be sampled, which would correspond to non-existing category.

Authors test out their proposed encoding method on several datasets in a TSTR manner (train on synthetic, test on real): they train a diffusion model to generate synthetic data, train a classifier/regressor on just generated data, and test the classifier/regressor on the actual data used for the diffusion model training. Results indicate that a proposed encoding is on par with simple log_2 encoding.

**Strengths:**

The proposed method is very simple to understand and implement.

**Weaknesses:**

Paper has two weaknesses: results and presentation
1. Results. On 5 tabular datasets where the such an encoding method would be of most use, the proposed is clearly better only on 2 of the tasks (CC, AR), whereas on BD and AD performance is on par, and I'm going to discount any results on IS due to the size of dataset (1338).  Similarly, when used for conditioning of GANs, visually speaking res-bit results seems to be worse (much less diverse) and have no strong edge over one-hot in classification tasks. Considering these observations, it is hard to say that residual bitwidth representation of categorical values is a good encoding in general.
2. Presentation & Motivation. I the writing and the flow of the paper hard to follow. Initial pages are more like a catalog book of ml methods (section 2 in particular) rather than a cohesive presentation of ideas. The paper has many stylistic issues like using "that this", "very widely used", "limit the increase in dimensionality to a logarithmic increase" and etc. Also, I find the motivation a bit underdeveloped. Section 3.1. explains the "out of index" issue but does not provide evidence whether it is indeed the main cause that limits the model performance. I, generally believe, that a well trained model would learn to not sample from category bits that does not exist. It would be an important addition to the paper to show not only better results but provide evidence that improvement was due to solving out-of-index issue.

**Questions:**

Please address weaknesses above as much as possible.

---

> ### Author Response · Authors · 2023-11-21
> **Response to reviewer Zunf (except reference)**
>
> Thank you for your constructive feedback. Here we response to your questions as much as possible.
>
> **As for Weakness 1**
>
> Results from the remaining datasets experimented in the TabDDPM paper, which include categorical columns, are presented in the COMMON RESPONSE. According to the findings, the results are comparable for the cardio dataset and slightly inferior for the abalone dataset. However, the facebook dataset, with 157,638 training data, demonstrates significant results. If we exclude abalone and IS due to the small dataset sizes, TRBD overall equals or surpasses TabDDPM. The above, coupled with the runtime results (refer to Section 4.1.5 and Table 4), are significant in that the same or better results can be achieved in a shorter time. Concerning the conditioning of GANs, when using CIFAR10, it can be visually said that more mode collapse can be observed in the one-hot case compared to ResBit (see Appendix. C). Furthermore, ResBit is superior to one-hot in terms of FID, so it is not a result that the superiority of ResBit is not generally recognized. As for image classification, there is certainly no strong advantage. However, as mentioned **twice** in our paper (see Sections 4.2 and 4.3), the main purpose of these two experiments is not to show the superiority in terms of accuracy. **We aim to confirm that ResBit can be used as an alternative to one-hot while reducing the number of dimensions (see Section 4.4)**. Therefore, the lack of strong significance in terms of accuracy is not a concern. In addition, even if it is somewhat inferior in terms of accuracy, in this experimental setting, the runtime has an advantage (refer to the table below). Considering the trade-off between runtime and quality, there is no compelling reason to strongly negate this advantage.
>
>
> Table 1. Comparison the runtime using SNGAN and 40 selected class from the Food-101.
> |  |  | | | |
> | :------: | :--------------------: | :--------------: | :--------------: | :--------------: |
> | Method  | Label dim | #params D | #params G | Training Time (hours)
> | one-hot | 40 | 38,446,736 | 32,806,147 | 8.75 |
> | ResBit | **9** | **38,424,896** | **32,298,243** | **8.25** |
>
>
> **As for Weakness 2**
>
> (Regarding motivation) Certainly a well-trained model may not generate data from non-existent category bits, but what makes it any different from over-fitting or copying training data? For example, large language models (LLMs) and text-to-image diffusion models (DMs) have been suggested to be able to extract training data [1, 2]. We consider tabular data to be more sensitive than images or text, since they often directly contain personal information, such as financial or physical data. In addition, the data used in our experiment is not large, 600k at most. Furthermore, the datasets used in recent table data generation using Diffusion Models [3, 4, 5] are no larger than this. In [6], it is stated that "Typically, this type of memorization happens when the model makes many passes over a small training set". It is thought that over-fitting is particularly likely to occur in the generation of tabular data, where small data sets are not uncommon. Therefore, it is not sufficient to train them on as much data as LLMs and DMs. Assuming it works, Machine Unlearning [7] is a way to address the privacy issue, but we believe it will only lead to loss of diversity in the generative task.

---

> > ### Author Response · Authors · 2023-11-21
> > **Response to reviewer Zunf (reference)**
> >
> > ## Reference
> > [1] Nicholas Carlini, Florian Tram`er, Eric Wallace, Matthew Jagielski, Ariel Herbert-Voss, Katherine Lee, Adam Roberts, Tom Brown, Dawn Song, Ulfar Erlingsson, Alina Oprea, and Colin Raffel. Extracting training data ´
> > from large language models. In 30th USENIX Security Symposium (USENIX Security 21), pages 2633–2650. USENIX Association, August 2021.
> >
> > [2] Nicolas Carlini, Jamie Hayes, Milad Nasr, Matthew Jagielski, Vikash Sehwag, Florian Tram`er, Borja Balle, Daphne Ippolito, and Eric Wallace. Extracting training data from diffusion models. In 32nd USENIX Security
> > Symposium (USENIX Security 23), pages 5253–5270, Anaheim, CA, August 2023. USENIX Association.
> >
> > [3] Chaejeong Lee, Jayoung Kim, and Noseong Park. CoDi: Co-evolving contrastive diffusion models for mixed-type tabular synthesis. In Andreas Krause, Emma Brunskill, Kyunghyun Cho, Barbara Engelhardt, Sivan Sabato, and Jonathan Scarlett (eds.), Proceedings of the 40th International Conference on Machine Learning, volume 202 of Proceedings of Machine Learning Research, pp. 18940–18956. PMLR, 23–29 Jul 2023. URL https://proceedings.mlr.press/v202/lee23i.html.
> >
> > [4] Jayoung Kim, Chaejeong Lee, and Noseong Park. STasy: Score-based tabular data synthesis. In The Eleventh International Conference on Learning Representations, 2023. URL https://openreview.net/forum?id=1mNssCWt_v.
> >
> > [5] Hengrui Zhang, Jiani Zhang, Balasubramaniam Srinivasan, Zhengyuan Shen, Xiao Qin, Christos Faloutsos, Huzefa Rangwala, and George Karypis. Mixed-type tabular data synthesis with score-based diffusion in latent space, 2023.
> >
> > [6] Aakanksha Chowdhery, Sharan Narang, Jacob Devlin, Maarten Bosma, Gaurav Mishra, Adam Roberts, Paul Barham, Hyung Won Chung, Charles Sutton, Sebastian Gehrmann, Parker Schuh, Kensen Shi, Sasha Tsvyashchenko, Joshua Maynez, Abhishek Rao, Parker Barnes, Yi Tay, Noam Shazeer, Vinodkumar Prab- hakaran, Emily Reif, Nan Du, Ben Hutchinson, Reiner Pope, James Bradbury, Jacob Austin, Michael Isard, Guy Gur-Ari, Pengcheng Yin, Toju Duke, Anselm Levskaya, Sanjay Ghemawat, Sunipa Dev, Henryk Michalewski, Xavier Garcia, Vedant Misra, Kevin Robinson, Liam Fedus, Denny Zhou, Daphne Ippolito, David Luan, Hyeon- taek Lim, Barret Zoph, Alexander Spiridonov, Ryan Sepassi, David Dohan, Shivani Agrawal, Mark Omernick, Andrew M. Dai, Thanumalayan Sankaranarayana Pillai, Marie Pellat, Aitor Lewkowycz, Erica Moreira, Rewon Child, Oleksandr Polozov, Katherine Lee, Zongwei Zhou, Xuezhi Wang, Brennan Saeta, Mark Diaz, Orhan Fi- rat, Michele Catasta, Jason Wei, Kathy Meier-Hellstern, Douglas Eck, Jeff Dean, Slav Petrov, and Noah Fiedel. Palm: Scaling language modeling with pathways, 2022.
> >
> > [7] Y. Cao and J. Yang, "Towards Making Systems Forget with Machine Unlearning," 2015 IEEE Symposium on Security and Privacy, San Jose, CA, USA, 2015, pp. 463-480, doi: 10.1109/SP.2015.35.

---

### Official Review · Reviewer_42Kk · 2023-11-01

**Soundness:** 1 poor
**Presentation:** 1 poor
**Contribution:** 1 poor
**Rating:** 3
**Confidence:** 4

**Summary:**

The paper proposes a hierarchical bit representation called Residual Bit Vector (ResBit) to address the complexity issue of one-hot encoding of categorical data. Because the number of elements of one-hot encoding grows linearly with the number of categories, the increased dimensionality may be harmful to performance. ResBit mainly follows the idea of residual vector quantization (Juang & Gray, 1982). It finds binary representation hierarchically and is shown to avoid the so-called “out-of-index” problem for some cases. Several experiments in tabular data generation, image generation, and image classification are conducted to study the performance of ResBit. Mixed results are reported.

**Strengths:**

I find it really hard to find the strengths of this paper. See the reasons below.

**Weaknesses:**

- There are several false claims in the paper. First, ResBit may not fully address the “out-of-index” issue. Since $N=50=32+16+2$, the example given in the paper is free from the issue. Proof for any natural number is missing. One can find a counterexample by find the ResBit representation of $N=51$? Second, the ResBit does not really improve or at least achieve no worse results compared to their baselines. In some cases, ResBit even performs much worse than the baselines.

- Some descriptions in the paper are not clear. For example, the authors claim that increasing the dimensionality can cause model learning to fail. It is not clear to me why and how it fails. For example, overparameterization can lead to better results. Providing some references could be helpful.

- In Section 4.1.4, the authors state that the loss exploded or disappeared during the training phase of TabDDPM for certain datasets and argue that that is probably due to the very large number of dimensions. This seems to be a strange reason because the dimensions are not too large in these problems and usually this kind of problem can be addressed by normalizing the features or using smaller learning rates.

- The runtime comparison seems unfair because the TabDDPM and TRBD use different networks with different number of layers.
In Section 4.3, it would make more sense to use ResBit for datasets like ImageNet. CIFAR-10 only has 10 classes so the reduction of the encoding of the categories is insignificant.

- In Section 4.4, the authors argue that ResBit reduces the representation complexity of categorical data. However, this would be only meaningful when the performance of ResBit is justified.

**Questions:**

1. Can we prove that ResBit does not have the “out-of-index” issue mathematically?

2. Given that ResBit is proposed for reducing the representation complexity of categorical data, have you tried to run ResBit for image classification on ImageNet? Does it maintain the performance compared to one-hot encoding while achieving lower complexity?

---

> ### Author Response · Authors · 2023-11-21
> **Response to reviewer 42Kk (except reference)**
>
> Thank you for your constructive comments. We response to your questions and concerns.
>
> > Can we prove that ResBit does not have the “out-of-index” issue mathematically?
>
> > First, ResBit may not fully address the “out-of-index” issue. Since $N=50=32+16+2$, the example given in the paper is free from the issue. Proof for any natural number is missing. One can find a counterexample by find the ResBit representation of $N=51$?
>
> Since ResBit is to be applied when the number of classes is known in advance, the reviewer seems to have a major misunderstanding. In the analog bits, "out of index" occurs when the length of a class increases, even if the number of classes is known in advance. The $N=50$ shown in our paper is a perfect example. If $N=2^k$, "out of index" does not occur. ResBit is an expression that depends on the number of classes, similar to one-hot, etc. In the example of the states of the United States, one-hot considers a 50-dimensional vector because one-hot depends on the number of classes.　In the following, we formulate the ResBit. The number of classes is set to $N\geq2$. When $N=1$, there is no problem in the actual task, so we do not consider it (e.g., if conditioning, it corresponds to no condition). Considering 0-index, ResBit can be written as follows.
>
> $$
>     N-1=\sum_{k=0}^{\infty}a_k(2^{k}-1)\qquad \mathrm{s.t.}\min\sum_{k=0}^{\infty}a_kk
> $$
>
> The $k$-th term can represent any natural number in the range $[0, a_k(2^k-1)]$. It is easy to decompose $i\in\mathbb{Z}$ with $0\leq i<N$ as the sum of such natural numbers.
>
>
> > Given that ResBit is proposed for reducing the representation complexity of categorical data, have you tried to run ResBit for image classification on ImageNet? Does it maintain the performance compared to one-hot encoding while achieving lower complexity?
>
> As described in Section 4.4, this is a future work. Note that there are two possible ways to evaluate the output of ResBit in this experiment. In both cases, the output of the model is rounded.
>
> - Check if the output of the model matches the ResBit of the correct label
> - Convert the model output to an integer value and check if it matches the correct label
>
> Since the accuracy was the same in both cases, the model is considered to correctly represent ResBit. Therefore, we expect the same degree of difficulty in experiments on datasets with numerous labels, such as ImageNet-1k, as in the one-hot case. Note that image classification in ImageNet is also possible by ensembling many classifiers like ACGAN[1].
>
>
> > the ResBit does not really improve or at least achieve no worse results compared to their baselines. In some cases, ResBit even performs much worse than the baselines.
>
> Many of our experiments show applicability. Learning with ResBit is equivalent to multi-label classification, and in some cases, devising multi-class classification alone may not be sufficient. In our experiments, the one-hot method is slightly more advantageous than the one-hot method because the one-hot classification employs top-1. However, this is not the case in most cases in the experiment. Therefore, only some bad results should not be emphasized.
>
>
> > The runtime comparison seems unfair because the TabDDPM and TRBD use different networks with different number of layers.
>
> As for the runtime comparison, we show in the table below the results when we use same networks with the same number of layers.
>
> |  |  | | |
> | :------: | :--------------------: | :--------------: | :--------------: |
> | Dataset  | layers | train | sample |
> | CH (TabDDPM) | [512, 1024, 1024, 1024] | 516s | 42s |
> | CH (TRBD) | [512, 1024, 1024, 1024] | **264s** | **21s** |
> | IS (TabDDPM) | [1024, 512, 512, 1024] | 385s | 28s |
> | IS (TRBD) | [1024, 512, 512, 1024] | **173s** | **17s** |
>
> Because the number of layers has been reduced, a slight speedup in training time has been achieved. On the other hand, TRBD's superiority in speed is unassailable; the difference in input dim is the difference between ResBit and one-hot.
>
> > In Section 4.1.4, the authors state that the loss exploded or disappeared during the training phase of TabDDPM for certain datasets and argue that that is probably due to the very large number of dimensions. This seems to be a strange reason because the dimensions are not too large in these problems and usually this kind of problem can be addressed by normalizing the features or using smaller learning rates.
>
> It is unfair if the experiment is not conducted in the same learning rate search space (fairness is what you pointed out as a weakness). In the same search space, the fact that the loss exploded or disappeared for TabDDPM but not for TRBD may be due to the number of dimensions.　This is because there is no other difference between TabDDPM and TRBD.　In the experiments with Multinomial Diffusion [2], the number of classes is 8, 27, and 256, so compared to those, 6k and 7k are very large and not something to consider in general terms.

---

> ### Author Response · Authors · 2023-11-21
> **Response to reviewer 42Kk (reference)**
>
> ## Reference
> [1] Augustus Odena and Christopher Olah and Jonathon Shlens. Conditional Image Synthesis with Auxiliary Classifier GANs. Proceedings of the 34th International Conference on Machine Learning, in Proceedings of Machine Learning Research 70:2642-2651 Available from https://proceedings.mlr.press/v70/odena17a.html.
>
> [2] Emiel Hoogeboom, Didrik Nielsen, Priyank Jaini, Patrick Forre ́, and Max Welling. Argmax flows and multinomial diffusion: Learning categorical distributions. In A. Beygelzimer, Y. Dauphin, P. Liang, and J. Wortman Vaughan (eds.), Advances in Neural Information Processing Systems, 2021. URL https://openreview.net/forum?id=6nbpPqUCIi7.

---

> > ### Comment · Reviewer_42Kk · 2023-11-22
> >
> > I express my gratitude to the authors for their responses. Several of my minor concerns have been appropriately addressed, leading me to revise my rating upward from 1 to 3. However, it is crucial to note that my significant reservations regarding ResBit remain unchanged. The assertions made about its performance and complexity reduction necessitate further experimentation and robust justifications.

---

### Official Review · Reviewer_Dspo · 2023-11-02

**Soundness:** 3 good
**Presentation:** 2 fair
**Contribution:** 2 fair
**Rating:** 5
**Confidence:** 2

**Summary:**

In the paper, the authors propose a Residual Bit Vector (ResBit), which is a hierarchical bit representation. Authors also show that such representation can be used to build a tabular data generation method called TRBD. TRBD can generate diverse and high-quality data from small-scale table data. ResBit was also used in GANs or conditioning.

**Strengths:**

1. The paper introduces the interesting extension of Analog Bits.
2. The paper has good theoretical fundaments.

**Weaknesses:**

1. In the abstract, the authors introduce methods in a different order than in the introduction. It is misleading. Maybe it is possible to do it consistently.
2. The first Fig 1. in the paper refers to the reference paper. Maybe at the beginning, authors can give some illustrations describing the new proposed method.
3. Some illustrations of the method should be added.
4. The model proposes three elements: ResBit, TRBD, and conditioning GAN. Unfortunately, none of such components are well evaluated. Especially ResBit should be compared with Analog Bits.
5. In TabDDPM, authors propose experiments on 15 datasets with many baselines. Authors should follow such an experimental setting.
6. Maybe authors should introduce fewer components but add more detailed comparisons with existing methods.
7. Maybe it is possible to run the algorithms on an image dataset.

**Questions:**

1. How the ResBit algorithm works concerning Analog Bits.
2. Maybe it is possible to show some practical tasks to show that ResBit works better than Analog Bits.
3. The United States example is convincing, but the authors should present that such a problem is a real problem in practical application.

---

> ### Author Response · Authors · 2023-11-21
> **Response to reviewer Dspo (except reference)**
>
> Thank you for your many valuable feedback and suggestions. Here we address responses to your questions and weakness.
>
> > How the ResBit algorithm works concerning Analog Bits.
>
> ResBit is a fusion of part of the Analog Bits concept, which represents discrete data as a sequence of bits, and the RVQ concept, which further vector quantizes the difference of vector quantization. The full application of the Analog Bits concept requires the high expressive power of Diffusion Models, and TRBD is an example of its application. On the other hand, ResBit can be considered to work the same as one-hot if it is considered as a method of expressing ResBit as  0/1. Its application to image classification and conditional image generation is a verification of this. In addition, if the number of classes is represented by $2^m$, it is truly the same as the analog bits. Representing the pixels of an image in such a way is exactly what was experimented in the analog bits paper.
>
> > Maybe it is possible to show some practical tasks to show that ResBit works better than Analog Bits
>
> > The model proposes three elements: ResBit, TRBD, and conditioning GAN. Unfortunately, none of such components are well evaluated. Especially ResBit should be compared with Analog Bits.
>
> We cannot evaluate in the tabular data generation task due to the "out of index" problem described in Section 3.1. On the other hand, we can evaluate the other two applications. Let $N$ be the number of classes and consider the smallest $m\in\mathbb{N}$ satisfying $N<2^m$ holds. In this experiment, $m$-dimensional vectors are used to represent the classes. For example, for $N=10$, $m=4$. The table below are the experimental results when using CIFAR-10. The results for one-hot and ResBit are those in the paper.
>
>
> |  |  | | |
> | :------: | :--------------------: | :--------------: | :--------------: |
> | Task  | Image Classification | Image Classification | Image Generation |
> | Used Model | MobileNetV3-small | MobileNetV3-large | InfoGAN |
> | Metrics | Accuracy (%) | Accuracy (%) | FID |
> | Result (one-hot) | 66.52 | **74.63** | 95.52 |
> | Result (ResBit) | **66.70** | 72.55 | 82.21 |
> | Result (analog) | 62.70 | 70.62 | **63.54** |
>
> In the context of conditioning, "out of index" scenario is not deemed a significant issue; rather, it leads to a straightforward reduction in the number of dimensions, which is believed to enhance accuracy. Conversely, in the realm of image classification, predicting an "out of index" situation could result in diminished accuracy. Consequently, we consider that Analog Bits are perceived as less versatile than ResBit.
>
> > The United States example is convincing, but the authors should present that such a problem is a real problem in practical application.
>
> There are much information exists in fraudulent transaction data in the financial field. In general, the ratio of anomaly data to total data is very low. In such cases, oversampling may be performed using methods such as SMOTE [1]. In recent years, deep learning-based methods have also been researched, in which the entire table data is generated once and then anomaly data is extracted after feature engineering and other processes are performed. The method of masking by special string [2] pointed out in our paper hides rare data, which may affect the accuracy of anomaly detection. Our method can generate all data without masking. If the accuracy of anomaly detection is not improved, it is easier to analyze whether the cause is the data generation method itself or the loss of diversity due to masking.
>
>
> > In TabDDPM, authors propose experiments on 15 datasets with many  baselines. Authors should follow such an experimental setting.
>
> > Maybe authors should introduce fewer components but add more detailed comparisons with existing methods.
>
> Of the 15 datasets listed in TabDDPM, the results of those not covered in the paper are shown in the COMMON RESPONSE. Note that we do not conduct the experiments for data that have only numerical datasets, since TRBD and TabDDPM are the same in such datasets. We use CatBoost in the TSTR framework. When integrated with the results in our paper, TRBD outperforms or equals TabDDPM on most datasets. This shows the superiority of the proposed method in generating tabular data. Also, since TabDDPM has been reported to significantly outperform existing deep generative models [3, 4, 5] in the TabDDPM paper, we do not believe that we need to use them as a baseline.
>
>
> > Maybe it is possible to run the algorithms on an image dataset.
>
> Representing the pixels of an image in ResBit is the same as representing them in Analog Bits (Since the pixels of an image are integer values of [0, 255]). An experiment to represent image pixels using Analog Bits and image generation using Diffusion Models has been conducted in [6]. There is no need for us to perform the experiments already validated in [6].

---

> ### Author Response · Authors · 2023-11-21
> **Response to reviewer Dspo (reference)**
>
> ## Reference
> [1] Nitesh V Chawla, Kevin W Bowyer, Lawrence O Hall, and W Philip Kegelmeyer. Smote: synthetic minority over-sampling technique. Journal of artificial intelli- gence research, 16:321–357, 2002.
>
> [2] Masane FUCHI, Amar ZANASHIR, Hiroto MINAMI, and Tomohiro TAKAGI. Generating a wide variety of categorical data using diffusion models. Proceedings of the Annual Conference of JSAI, JSAI2023:2K5GS203–2K5GS203, 2023. doi: 10.11517/pjsai.JSAI2023.0 2K5GS203.
>
> [3] Zilong Zhao, Aditya Kunar, Robert Birke, and Lydia Y Chen. Ctab-gan: Effective table data synthesizing. In Asian Conference on Machine Learning, pp. 97–112. PMLR, 2021.
>
> [4] Lei Xu, Maria Skoularidou, Alfredo Cuesta-Infante, and Kalyan Veeramachaneni. Modeling tabular data using conditional gan. In Advances in Neural Information Processing Systems, 2019.
>
> [5]  Zilong Zhao, Aditya Kunar, Robert Birke, and Lydia Y. Chen. Ctab-gan+: Enhancing tabular data synthesis, 2022.
>
> [6] Ting Chen, Ruixiang ZHANG, and Geoffrey Hinton. Analog bits: Generating discrete data using diffusion models with self-conditioning. In The Eleventh International Conference on Learning Representations, 2023. URL https://openreview.net/forum?id=3itjR9QxFw.

---

### Author Response · Authors · 2023-11-21
**COMMON RESPONSE**

Of the 15 datasets listed in TabDDPM, the results of those not covered in our paper are shown below table. We use CatBoost as a classifier in the TSTR framework.


|  |  | | |
| :------: | :--------------------: | :--------------: | :--------------: |
| Dataset  | Method | F1 ($R^2$) | AUROC (RMSE) |
| Cardio | Identity | $0.738\pm0.001$ | $0.805\pm0.000$ |
| Cardio |  TabDDPM | $\boldsymbol{0.737\pm0.001}$ | $\boldsymbol{0.803\pm0.001}$ |
| Cardio | TRBD (ours) | $\boldsymbol{0.737\pm0.002}$ |  $\boldsymbol{0.803\pm0.001}$ |
|||||
| King | Identity | $0.906\pm0.002$ | $107,633.528\pm1,022.334$ |
| King | TabDDPM | $0.864\pm0.112$ | $129,491.185\pm5,426.840$ |
| King | TRBD (ours) |  $\boldsymbol{0.892\pm0.004}$ | $\boldsymbol{115,411.676\pm1,972.408}$ |
|||||
| Abalone | Identity | $0.57\pm0.002$ | $2.009\pm0.004$ |
| Abalone | TabDDPM | $\boldsymbol{0.567\pm0.012}$ | $\boldsymbol{1.986\pm0.028}$ |
| Abalone | TRBD (ours) |   $0.560\pm0.006$ |  $2.002\pm0.013$ |
|||||
| Facebook | Identity | $0.838\pm0.001$ | $5.297\pm0.019$ |
| Facebook | TabDDPM | $0.662\pm0.008$ | $7.648\pm0.084$ |
| Facebook | TRBD (ours) |   $\boldsymbol{0.706\pm0.005}$ |  $\boldsymbol{7.127\pm0.057}$ |
|||||
| Churn Modeling | Identity | $0.580\pm0.033$ | $0.855\pm0.010$ |
| Churn Modeling | TabDDPM | $0.601\pm0.007$ | $\boldsymbol{0.873\pm0.002}$ |
| Churn Modeling | TRBD (ours) | $\boldsymbol{0.612\pm0.009}$ | $\boldsymbol{0.870\pm0.002}$

---

### Meta-Review · Area_Chair_z3AP · 2023-12-08

**Metareview:**

The paper introduces Residual Bit Vector (ResBit) to tackle challenges in one-hot encoding of categorical data and presents the tabular data generation method TRBD. It is rejected due to concerns raised by all reviewers regarding clarity, evaluation shortcomings, and inadequate motivation.

**Justification For Why Not Higher Score:**

Reviewers highlight issues with the ordering of methods, call for clearer illustrations, and express concerns about the evaluation and comparison of ResBit with baselines. Inconsistencies and a lack of cohesive flow in the initial sections also contribute to the rejection.

**Justification For Why Not Lower Score:**

N/A

---

### Decision · Program_Chairs · 2024-01-16

Reject